# Environmental Pollution Indices and Multivariate Modeling Approaches for Assessing the Potentially Harmful Elements in Bottom Sediments of Qaroun Lake, Egypt

Ali H. Saleh [1], Mohamed Gad [2], Moataz M. Khalifa [3,4], Salah Elsayed [5,*], Farahat S. Moghanm [6], Adel M. Ghoneim [7], Subhan Danish [8,*], Rahul Datta [9,*], Moustapha E. Moustapha [10] and Magda M. Abou El-Safa [1]

1   Environmental Geology, Surveying of Natural Resources in Environmental Systems Department, Environmental Studies and Research Institute, University of Sadat City, Sadat City 32897, Minufiya, Egypt; ali.saleh@esri.usc.edu.eg (A.H.S.); magda.aboelsafa@esri.usc.edu.eg (M.M.A.E.-S.)
2   Hydrogeology, Evaluation of Natural Resources Department, Environmental Studies and Research Institute, University of Sadat City, Sadat City 32897, Minufiya, Egypt; mohamed.gad@esri.usc.edu.eg
3   Geology Department, Faculty of Science, Institute of Earth Sciences, Southern Federal University, Zorge St., 40, 344090 Rostov-on-Don, Russia; moutaz.khalifa@science.menofia.edu.eg
4   Geology Department, Faculty of Science, Menoufia University, Shiben El Kom 51123, Minufiya, Egypt
5   Agricultural Engineering, Evaluation of Natural Resources Department, Environmental Studies and Research Institute, University of Sadat City, Sadat City 32897, Minufiya, Egypt
6   Soil and Water Department, Faculty of Agriculture, Kafrelsheikh University, Kafr El-Sheikh 33516, Kafr El Sheikh, Egypt; fsaadr@yahoo.ca
7   Agricultural Research Center, Field Crops Research Institute, Giza 12112, Giza, Egypt; adelrrtc.ghoneim@gmail.com
8   Department of Soil Science, Faculty of Agricultural Sciences and Technology, Bahauddin Zakariya University, Multan 60800, Pakistan
9   Department of Geology and Pedology, Faculty of Forestry and Wood Technology, Mendel University in Brno, Zemedelska1, 61300 Brno, Czech Republic
10  Department of Chemistry, College of Science and Humanities, Prince Sattam bin Abdulaziz University, Al-Kharj 11942, Saudi Arabia; m.moustapha@psau.edu.sa
*   Correspondence: salah.emam@esri.usc.edu.eg (S.E.); sd96850@gmail.com (S.D.); rahulmedcure@gmail.com (R.D.)

**Abstract:** This research intends to offer a scientific foundation for environmental monitoring and early warning which will aid in the environmental protection management of Qaroun Lake. Qaroun Lake is increasingly influenced by untreated wastewater discharge from many anthropogenic activities, making it vulnerable to pollution. For that, six environmental pollution indices, namely contamination factor (Cf), enrichment factor (EF), geo-accumulation index (Igeo), degree of contamination (Dc), pollution load index (PLI), and potential ecological risk index (RI), were utilized to assess the bottom sediment and to determine the different geo-environmental variables affecting the lake system. Cluster analysis (CA), and principal component analysis (PCA) were used to explore the potential pollution sources of heavy metal. Moreover, the efficiency of partial least-square regression (PLSR) and multiple linear regression (MLR) were tested to assess the Dc, PLI, and RI depending on the selected elements. The sediment samples were carefully collected from 16 locations of Qaroun Lake in two investigated years in 2018 and 2019. Total concentrations of Al, As, Ba, Cd, Co, Cr, Cu, Fe, Ga, Hf, Li, Mg, Mn, Mo, Ni, P, Pb, Sb, Se, Zn, and Zr were quantified using inductively coupled plasma mass spectra (ICP-MS). According to the Cf, EF, and $I_{geo}$ results, As, Cd, Ga, Hf, P, Sb, Se, and Zr demonstrated significant enrichment in sediment and were derived from anthropogenic sources. According to Dc results, all collected samples were categorized under a very high degree of contamination. Further, the results of RI showed that the lake is at very high ecological risk. Meanwhile, the PLI data indicated 59% of lake was polluted and 41% had PLI < 1. The PLSR and MLR models based on studied elements presented the highest efficiency as alternative approaches to assess the Dc, PLI, and RI of sediments. For examples, the validation (Val.) models presented the best performance of these indices, with $R^2$val = 0.948–0.989 and with model accuracy ACCv = 0.984–0.999 for PLSR, and with $R^2$val = 0.760–0.979 and with ACCv = 0.867–0.984 for MLR. Both models for Dc,

PLI, and RI showed that there was no clear overfitting or underfitting between measuring, calibrating, and validating datasets. Finally, the combinations of Cf, EF, I$_{geo}$, PLI, Dc, RI, CA, PCA, PLSR, and MLR approaches represent valuable and applicable methods for assessing the risk of potentially harmful elemental contamination in the sediment of Qaroun Lake.

**Keywords:** Pollution load index; potential ecological risk index; degree of contamination; enrichment factor; contamination factor; geoaccumulation index; heavy metals; PLSR; MLR

## 1. Introduction

The assessment of lake environments has become one of the most global topics to have received a lot of attention from researchers, especially in arid and semi-arid regions [1], because they are vital to climate management, social-economic activity, and environmental preservation [2–4]. The contamination level in lakes reflects environmental pollution, which results from industry, agriculture, and unplanned urbanization. Thus, the continuous assessment of the heavy metals in the aquatic environment in Lakes is important [5–9], because heavy metals are characterized by toxicity, accumulation in the food chain, environmental sensitivity, and being non-biodegradable [5–9].

Qaroun Lake is a natural inland saline lake and represents one of the main geomorphological features of the Egyptian Western Desert. It is a place for fisheries, salt production, tourism, and migratory birds in the Autumn and Winter seasons, in addition to being a natural discharge area for El-Fayoum province [10–12]. The Qaroun Lake area distinguishes itself by its richness in terms of biodiversity, geological formations, and archaeological sites. Furthermore, the lake is a globally significant wetland for Autumn and Wintering migratory water birds. Consequently, in 1989, it was designated as a natural protectorate according to Law 102/1983 by Prime Ministerial Decree No. 943/1989 [13]. In addition, it was originally a freshwater lake and changed into a saltwater habitat [14]. El-Fayoum province discharges a massive amount (more than 450 million m$^3$/year) of untreated agricultural, industrial, aquacultural, sewage, and domestic effluents into the Qaroun Lake [15–21]. The lake has no surface outlet and loses its water through evaporation processes only. This causes a gradual increase of lake water salinity and their pollutant content, depending on the evaporation rate and the quantity of drainage water inflow [15,19,22]. As a result of the accumulation of pollutants, the environmental quality of the Qaroun Lake will change and affect the food chain and ecosystem [23,24]. Consequently, economic activities such as fishing, and tourism will stop. Moreover, migratory birds will be affected by the lake's environmental degradation. The Qaroun Lake is subject to deterioration, as evidenced by previous studies carried out by El-Kady et al. [12]; Attia et al. [11]; Redwan and Elhaddad [25]; Abdel Wahed et al. [18]; El-Sayed et al. [10]; Soliman et al. [24]; Barakat et al. [17]; Abdel-Satar et al. [14].

The severe contamination of sediment by metals mixture can lead to the extinction of aquatic life [26]. Natural and anthropogenic resources contributed to the release of metals into the environment. Anthropogenic metals in sediment have higher mobility, bioavailability, and deleterious effects on aquatic organisms than lithogenic metals [27,28]. Sediment quality evaluations and aquatic environmental protection require an interpretation of the spatial distribution of metals in bottom sediment, inferring the possible ecological risk [10,29]. In the aquatic environment, the metal contaminants are absorbed and settled into sediments by particulate matter. Then, the pollutants return into the waters via the desorption mechanism [6,30]. Sediments constitute an important part of the aquatic ecosystem because they promote biodiversity, provide a habitat for numerous benthic creatures, and help to preserve water quality [31,32]. In addition, sediments meticulously record evidence related to human activity and serve as important factors for assessing pollution sources, history, dispersion, and environmental risk [33–36]. Sediments recognize as a critical indicator for pollutants monitoring in aquatic ecosystems [37–39]. Sediment analysis is chosen for researching metals for two main reasons [40]. (1) The metal concentra-

tions in sediment are higher than that of the linked water body, which improves the test's accuracy and validity [41]. (2) Regardless of environmental changes, metal accumulation in sediment is relatively constant [30,42]. The distribution of metals in sediments adjacent to the populated areas could be applied to study their effects on ecosystems and evaluate the environmental risks caused by waste discharged [12,43,44].

The study of metal concentrations in sediment assists environmental managers in better understanding how elements behave in aquatic ecosystems. It also aids in the evaluation of metal pollution in sediments which helps them to monitor water quality [44,45]. So, several geochemical and statistical methods are established to assess the quality of the aquatic ecosystem and predict their sustainability by evaluating the environmental risk of metals in surface sediments based on total concentration and toxicity [46–48]. For single-element contamination evaluation, the CF, EF, and $I_{geo}$ are commonly used. The Dc, PLI, and RI have been created to assess the combined danger of several elements in sediments [49,50]. The integration of these assessment approaches can effectively improve the precision of heavy metal contamination evaluations in surface sediments [44,51].

The multivariate modelling processing of environmental data, e.g., principal component analysis (PCA) and cluster analysis (CA), is commonly applied to identify potential pollution sources that affect aquatic systems. It represents an effective approach to natural resource management and assists in selecting the best solutions to pollution problems [52–54]. The PCA and CA are used to classify metals or investigated parameters into distinct factors/groups based on the predicted source of contribution and can assist in the organization and simplification of huge data sets to provide useful insight [55].

Calculating the PLI, RI, and Dc necessitates a series of calculations that demand a long time and great effort to transform numerous numbers from the sediment's metals data into a single value that describes the level of contamination. For that, both PLSR and MLR methods were used in this study to solve this issue. They are common approaches to express a linear relationship between independent and dependent variables [56]. They can combine data from many metals into a single index to improve the accuracy of a measured variable's prediction. Moreover, for resolving substantially multicollinear and noisy variables and efficiently assessing observed parameters, PLSR has been proposed [57]. Many collinear components can be reduced to a few non-correlated latent factors to minimize data overfitting or underfitting and reduce redundant data using PLSR [58,59]. Based on the advantages of these methods, the PLI, RI, and Dc and other pollution indices can be simultaneously computed from numerous heavy data using these methods. To the best of our knowledge, few studies have compared the performance of PLSR and MLR in predicting pollution indices using metals data.

Therefore, the objectives of this research were to (i) recognize the current situation of heavy metal concentrations and their spatial distributions in Qaroun Lake surface sediments; (ii) estimate the pollution level and environmental risks of heavy metals in the surface sediment of Qaroun Lake by calculating contamination factor (Cf), enrichment factor (EF), geo-accumulation index ($I_{geo}$), degree of contamination (Dc), pollution load index (PLI), and potential ecological risk index (RI); (ii) explore the potential pollution sources of heavy metals by using CA and PCA techniques; and (iv) evaluate the performance of PLSR and MLR models based on investigated potentially harmful elements to predict the three pollution indices, PLI, RI, and Dc.

## 2. Material and Methods

### 2.1. Study Area

Geographically, Qaroun Lake is sited in the northern deepest part (43 m below sea level) of the El Fayoum Depression, located in the Western Desert of Egypt, about 95 km southwest of Cairo. It exists between longitudes of 30°24′ and 30°50′ E and latitudes 29°24′ and 29°33′ N (Figure 1). Qaroun Lake is an elongated rectangular inland saline lake, about 45 km length from east to west and 5.7 km width from south to north, and its water depth ranges from 1 to 8.8 m. Only 18% of the lake area has a water depth greater than 5 m, while

over 67% of the lake area falls between the 2 and 5 m depth contours [60]. Qaroun Lake covers a surface area of 235 km$^2$ and it has a volume of water 1,100,000,000 m$^3$ [16,61]. It is surrounded to the east and south by several human activities and land uses, such as agricultural (represent area 28.20% of El-Fayoum province) and industrial activities (Kom Oshim industrial zone, Amisal salt production company). Further, urbanized areas (covered an area 4.07% of El-Fayoum province), entertainment activities (tourist resorts), aquaculture, and a dense roads network [20]. It is the third-largest lake in Egypt. It is distinguished by the presence of El Qarn El Zahabi Island, which is located in the middle of the lake, covers 1.5 km$^2$, and is a popular nesting spot for birds. The lake's water comes directly from the El-Fayoum province's agricultural drainage system. Another indirect water source comes from the groundwater seepage from the surrounding cultivated land. Qaroun lake receives its water primarily from two drains, namely El-Bats and El-Wadi, providing about 338 million m$^3$/year [15,16,19,20]. Further, in the southern regions of the lake, a total of 12 secondary drains discharge into it (Figure 1).

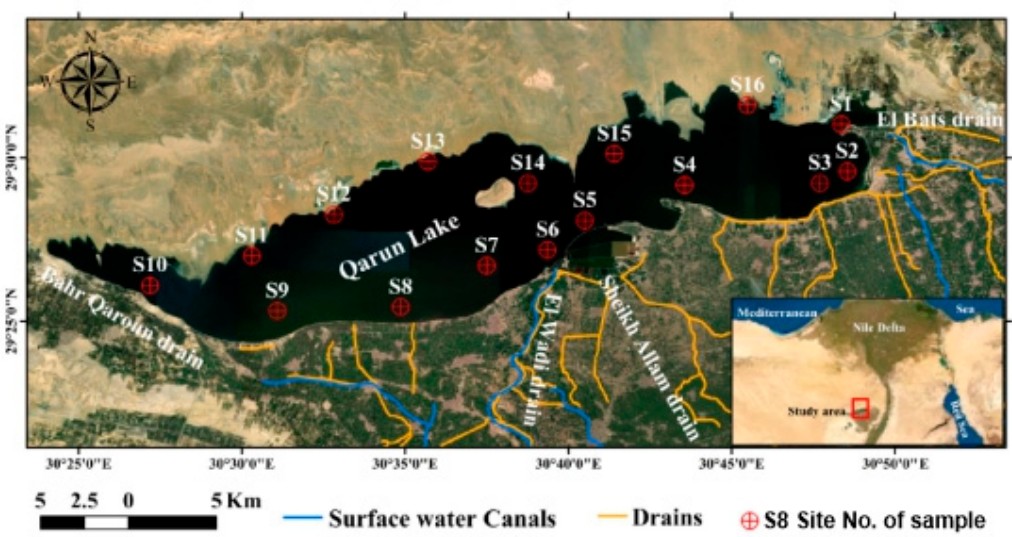

**Figure 1.** Location map of the study area and gauging sites.

Geologically, in the Fayoum area, the stratigraphic succession extends from Quaternary to Tertiary. The Fayoum Depression excavate in Middle Eocene rocks, which composed of gyps-ferrous shale, white marls, limestone, and sandstone [62]. Quaternary deposits formed of eolian, Nilotic (alluvial sediments), and lacustrine deposits surround the Qaroun Lake. Sands and gravels of various sizes intercalated with calcareous silt and clay content in alluvial sediments. Claystone, gypsum, and calcareous minerals intercalated with ferruginous sandy silt in the lacustrine deposits [63]. The bottom sediments of eastern and southern sides of the Qaroun Lake composed fine fractions (less than 63 microns) of about > 70%. The organic matter in the lake bottom sediment ranged between 3 and 17, the lake water salinity varied from 28 g/L to 40 g/L, and surface temperature ranged between 22 to 25 °C, while the drain water salinity was recorded to be 0.6 to 2 g/L. The sand dunes composed more than 1% of heavy minerals [19,21,24].

### 2.2. Sampling and Analyses

The representative bottom sediment samples of Qaroun Lake were collected using a Van Veen Grab sampler (US EPA, [64]) in the autumn of two investigated years in 2018 and 2019 from 16 sites (three samples at each site) (Figure 1). The collected samples were obtained from depths ranging between 2 and 5 m. The geographical information of each sampling location was identified using a hand-held Global Positioning System (GPS) device (Garmin/eTrex Vista HCx/personal navigator). The obtained sediment samples were promptly sealed in plastic bags and transferred to the laboratory in an ice-filled

field box. In the laboratory, sediment samples were air-dried to reach the constant weight before being sieved through 2.0 mm sieves. It was then ground with an agate mechanical mill (Retsch RM200) and stored in glass bottles to use in geochemical studies. Data were referred to the sediments of the eastern and southern sides of the Lake with fine fractions of about >70% less than 63 microns.

Sediment samples were acid-digested according to the US EPA 3052 method [65], using the Speedwave microwave digestion system. Deionized water, concentrated nitric acid, and concentrated hydrofluoric acid were utilized as reagents. The volume was diluted to 50 mL in the end. The current study adopted quality control methods, such as replicating samples or reference material and employing standard sediment reference materials (GBW07333) provided by China's Second Institute of Oceanography (SOA). Experiment vessel contamination was avoided by pre-cleaning and immersing the used vessel for at least 24 h in dilute nitric acid, followed by soaking and rinsing with deionized water. For precision testing, blanks were prepared in the same way and with the same reagents. Additionally, before analysis, the daily performance of the ICP-MS was examined to ensure that the machine was operating according to the manufacturer's specifications.

The concentration of elements Al, As, Ba, Cd, Co, Cr, Cu, Fe, Ga, Hf, Li, Mg, Mn, Mo, Ni, P, Pb, Sb, Se, Zn, and Zr were quantified according to inductively coupled plasma mass spectra (ICAP TQ ICP-MS Thermo Fisher Scientific Inc., Waltham, MA, USA). The laboratory analyses were carried out at the Environmental Geology Lab (EGL) and Environmental and Food Lab (EFL), University of Sadat City (ISO/IEC 17025/2017).

*2.3. Environmental Pollution Indices*

The anthropogenic toxic metals contamination in bottom surface sediments of Qaroun lake were assessed by applying ecological risk assessment approaches as Cf, Dc, EF, $I_{geo}$, PLI, and RI.

2.3.1. Contamination Factor and Degree of Contamination

According to Håkanson [66], the *Cf* and *Dc* are determined based on metal concentration readings in sediment. A contamination factor is an excellent approach for monitoring sediment pollution [67]. The *Cf* was calculated by dividing the concentration of each element in sediment by the background concentration (Equation (1)). Dc is another *Cf*-based index that can be defined as the total of *Cfs* for a specific place (Equation (2)).

$$Cf = \frac{M_x}{M_b} \tag{1}$$

$$Dc = \sum_{i=1}^{i=n} Cf \tag{2}$$

where $M_x$ is the metal concentration value in the investigated sediment, $M_b$ is the preindustrial reference value of the same metal (geochemical background value), and *n* is the number of the investigated elements in the sediment sample (in the present work *n* = 21). Since there were no local geochemical background values established for the metal abundances of the upper continental crust, Taylor et al. [68] were considered for their geochemical background. The descriptive terminology of the contamination factor and the degree of contamination are listed in Table 1.

2.3.2. Enrichment Factor

The EF is a successful tool to assess the impact of anthropogenic activities in sediments [69–71]. The EF values were calculated using the following Equation (3).

$$EF_x = \frac{(C_x/C_{Al})\ \text{sample}}{(C_x/C_{Al})\ \text{background}} \tag{3}$$

where $(C_x/C_{Al})$ sample is the ratio of measured metal to Al in the tested sample and $(C_x/C_{Al})$ background is the ratio value of the metal to Al in the natural geochemical back-

ground. The anthropogenic contamination forecast for the tested element improves by applying normalization against the tested element background value in the enrichment factor [72]. The aluminium (Al), iron (Fe), manganese (Mn), and titanium (Ti) are the common elements used as normalized elements in the enrichment factor calculation [28,73,74]. In the present study, Al was considered as a normalization element since the mobility of Al was very small [67]. Seven contamination classes were generally recognized according to the value of enrichment factors (Table 1).

**Table 1.** Classes of contamination factor (Cf), degree of contamination (Dc), enrichment factor (EF), geo-accumulation index (I_geo), pollution load index (PLI), and potential ecological risk index (RI).

| Contamination Indices | Indices Values | Classes | Reference |
|---|---|---|---|
| Cf | $\leq 1$<br>$1 < CF \leq 3$<br>$3 < CF \leq 6$<br>$6 < CF$ | Low<br>Moderate<br>Considerable<br>Very high | [66] |
| Dc | Dc < 8<br>8 < Dc < 16<br>16 < Dc < 32<br>Dc > 32 | Low<br>Moderate<br>Considerable<br>Very high | [66] |
| EF | <1<br>1–3<br>3–5<br>5–10<br>10–25<br>25–50<br>>50 | No enrichment<br>Minor enrichment<br>Moderate enrichment<br>Moderately severe enrichment<br>Severe enrichment<br>Very severe enrichment<br>Extremely severe enrichment | [75] |
| I_geo | $I_{geo} \leq 0$<br>$0 < I_{geo} \leq 1$<br>$1 < I_{geo} \leq 2$<br>$2 < I_{geo} \leq 3$<br>$3 < I_{geo} \leq 4$<br>$4 < I_{geo} \leq 5$<br>$5 < I_{geo}$ | Unpolluted<br>Unpolluted to moderately polluted<br>Moderately polluted<br>Moderately to strongly polluted<br>Strongly polluted<br>Strongly to extremely polluted<br>Extremely polluted | [76,77] |
| PLI | 1 > PLI<br>1 < PLI | Unpolluted<br>Polluted | [78] |
| RI | RI < 150<br>150 < RI < 300<br>300 < RI < 600<br>600 < RI | Low ecological risk<br>Moderate ecological risk<br>Considerable ecological risk<br>Very high ecological risk | [66] |

2.3.3. Geo-Accumulation Index

The I_geo is widely used for assessing element contamination in sediment [8,79]. Müller [76] proposed the following Equation (4) to calculate the I_geo, which is used to assess metal contamination levels in sediment samples.

$$I_{geo} = \log_2\left(\frac{Cn}{1.5Bn}\right) \quad (4)$$

where *Cn* represents the concentration of heavy metal in the sediment that has been measured. *Bn* is an element's geochemical background. For possible differences in the background data due to lithological variations, a factor of 1.5 was applied. The I_geo was divided into seven categories listed in Table 1.

### 2.3.4. Pollution Load Index

The PLI is one of the most effective approaches for evaluating sediment pollution status. It presents a cumulative indication of a sample's overall level of heavy metal pollution [69,76]. It was calculated for each sediment sampling site of Qaroun lake according to the following Equation (5) introduced by Tomlinson et al. [80].

$$PLI = (Cf_1 \times Cf_2 \times Cf_3 \times \ldots \times Cf_n)^{1/n} \tag{5}$$

where Cf is the contamination factor for measured metals in sediment samples; n is the number of metals tested in each sample (n = 21). The PLI for Qaroun Lake was calculated as the nth root of the multiplications of PLI for each sediment sampling site (Equation. (6)). Based on the PLI results, the investigated sediments can be divided into two groups: nonpolluted (PLI < 1) and polluted (PLI > 1) (Table 1).

$$PLI = (PLI_1 \times PLI_2 \times PLI_3 \times \ldots \times PLI_n)^{1/n} \tag{6}$$

### 2.3.5. Potential Ecological Risk Index

The RI was created by Håkanson [66] and provided in the following Equations (7) and (8). It was used to quantify the ecological risk of heavy metals in sediment [20,81]. The RI assesses the potential ecological danger posed by heavy metals in sediment by representing the susceptibility of various biological populations to hazardous chemicals [43,82].

$$RI = \sum_1^n Er \tag{7}$$

$$Er = Tr \times Cf \tag{8}$$

where Er is the potential ecological risk factor of the individual element and Tr is the toxic response factor suggested by Håkanson [66] for the metals. The RI is classified into four categories, ranging from low to very high ecological risk, as shown in Table 1.

### 2.4. Data Analysis

Statistical analysis was utilized to appraise the complex eco-toxicological processes. Hence, it showed the relationship and interdependence among the variables and their respective weights [83]. The data results of metals in the Qaroun lake sediment samples were processed by implementing the descriptive statistical parameters such as minimum (min), maximum (max), mean and standard deviation (SD). The statistical analyses of the data were carried out using PAST 4.07 (Natural History Museum, University of Oslo). The maps were created with GIS methodology version 10.2.1. The Pearson correlation coefficient was used to identify correlations between the geochemical properties of sediment samples [84–86]. Moreover, the significance thresholds were detected at 0.05 and 0.001 of *p*-value. Additionally, their relationships and likely causes were evaluated by using multivariate statistical methods such as principal component analysis (PCA) and cluster analysis (CA). CA (Ward's method) was used to assess the similarities and differences between sampling sites in terms of metal concentration [84]. CA and PCA were used to find a likely source of metals as well as factors that influence their concentration and spatial dispersion the most [87–89].

### 2.5. Partial Least-Square Regression (PLSR) and Multiple Linear Regression (MLR)

The PLSR and MLR models were tested in this study as new approaches to assess the three pollution indices (PLI, RI and Dc). Both models were created using version 10.2 of the unscramble X program (CAMO Software AS, Oslo, Norway). The PLSR and MLR model included the investigated 21 elements in this study as input variables (independent variables) to predict the PLI, RI, and Dc as output variables (dependent variables).

PLSR was used in conjunction with leave-one-out cross-validation (LOOCV) to link the input variables to the output variables. Choosing the optimal number of latent variables

(LVs) to represent the calibration data without overfitting or underfitting is an important step in PLSR analysis. To improve the robustness of the results, random 10-fold cross-validation was applied to the datasets.

For the MLR, the regression was used to calculate the parameters using the least-square approach, which minimized the sum of the errors squared.

The performance of the PLSR and MLR in predicting the three pollution indices for calibration (Cal.) and validation (Val.) models were assessed using four criteria: (1) $R^2$ coefficient; (2) root mean square error (RMSE); (3) mean absolute deviation (MAD); (4) Accuracy (ACC) of the models. The best models were selected based on the least RMSE and MAD as well as the highest $R^2$ and Acc. The $R^2$, RMSE, MAD, and ACC were calculated using Equations (9)–(12), respectively.

$$R^2 = 1 - \frac{\sum_{i=1}^{n}(\text{PIso}_i - \text{PIs}_{fi})^2}{\sum_{i=1}^{n}(\text{PIso}_i)^2} \tag{9}$$

$$\text{RMSE} = \sqrt{\frac{\sum_{i=1}^{n}(\text{PIso}_i - \text{PIs}_{fi})^2}{n}} \tag{10}$$

$$\text{MAD} = \frac{\sum_{i=1}^{n}|\text{PIso}_i - \text{PIs}_{fi}|}{n} \tag{11}$$

$$\text{Acc} = 1 - \text{ abs}\left(\text{mean} \frac{\text{PIs}_{fi - \text{PIso}_i}}{\text{PIso}_i}\right) \tag{12}$$

The measured value is PIsoi, the number of data points is n, and the predicted value is $\text{PIs}_{fi}$.

## 3. Results

### 3.1. Distribution of Elements

The metal concentrations of sediment samples were collected from Qaroun Lake in two investigated years, 2018 and 2019, in terms of statistical description, as shown in Table 2. Among the twenty-one examined metals, the mean concentrations values in the sediment samples increased in the order of Mo, Se, Pb, Cd, Sb, Hf, Co, Ni, As, Li, Cu, Zn, Cr, Ga, Ba, Mn, Mg, P, Zr, Fe, and Al (Table 2). Tables S1 and S2 showed the average concentration values of metals in Qaroun Lake's bottom sediment over two years. The Al and Fe concentration values decreased in the collected samples from the western and northwest sides of the lake (sites no. 10, 11, and 12) and increased in the eastern side sediments. The mentioned metals recorded the highest percentage in site no. 8. Zirconium had a distribution trend like aluminium and iron. It ranged from 0.21% in sample no. 10 to 1.23% in site no. 3. The sediment samples of the eastern part of Qaroun Lake had recorded the highest concentration values of Cd, Co, Cr, Cu, Ni, Pb, Zn, Mn, Hf, and Ga. However, the lowest concentration values of P, Se, and Mg were listed in the sediment of the eastern side of the Lake. Except for Se, the samples collected from the western part of the lake contain the minimum concentration values of Cd, Co, Cr, Cu, Pb, Zn, Sb, Mn, Hf, and Ba. The middle part sediment samples of the lake indicated the maximum concentration values of metals such as P, Sb, As, Mg, Ba, Li, and Mo. The comparison of results for the two investigated years, 2018 and 2019, showed a non-significant change in the average concentration of Al and Fe and a noticeable increase in the average concentration of As, Ba, Cd, Cr, Cu, Ga, Hf, Li, Mn, Sb, P, Pb, Se, and Zr. Meanwhile, Co, Mg, Mo, Ni, and Zn average concentrations recorded decreasing average values.

**Table 2.** Statistical description of metal concentrations in ppm, except for aluminium, iron, and zirconium which are in %.

| | **Al** | **As** | **Ba** | **Cd** | **Co** | **Cr** | **Cu** | **Fe** | **Ga** | **Hf** | **Li** |
|---|---|---|---|---|---|---|---|---|---|---|---|
| | | | | | **Metal Concentrations Values** | | | | | | |
| | | | | | First year 2018 ($n$ = 48) | | | | | | |
| Min | 0.34 | 3.37 | 55.4 | 3.23 | 2.05 | 6.07 | 3.01 | 0.23 | 28.1 | 3.92 | 10.3 |
| Max | 2.17 | 33.5 | 295 | 7.81 | 19.6 | 74.3 | 69.1 | 1.82 | 290 | 20.0 | 40.0 |
| Mean | 1.31 | 18.3 | 167 | 5.12 | 11.6 | 34.6 | 22.4 | 0.89 | 152 | 9.08 | 20.1 |
| SD | 0.49 | 8.21 | 74.8 | 1.10 | 5.33 | 19.0 | 16.0 | 0.49 | 67.7 | 4.11 | 6.85 |
| | | | | | Second year 2019 ($n$ = 48) | | | | | | |
| Min | 0.31 | 4.59 | 45.6 | 3.33 | 0.79 | 6.55 | 3.60 | 0.21 | 36.3 | 3.73 | 12.9 |
| Max | 2.13 | 36.5 | 286 | 7.38 | 22.5 | 72.2 | 68.3 | 1.89 | 324 | 18.4 | 38.1 |
| Mean | 1.30 | 18.8 | 168 | 5.25 | 11.5 | 34.7 | 22.5 | 0.89 | 156 | 9.55 | 20.1 |
| SD | 0.51 | 8.87 | 78.3 | 1.02 | 5.08 | 18.9 | 16.0 | 0.48 | 71.0 | 4.02 | 6.72 |
| | | | | | Data across two years ($n$ = 96) | | | | | | |
| Min | 0.31 | 3.37 | 45.6 | 3.23 | 0.79 | 6.07 | 3.01 | 0.21 | 28.1 | 3.73 | 10.3 |
| Max | 2.17 | 36.5 | 295 | 7.81 | 22.5 | 74.3 | 69.1 | 1.89 | 324 | 20.01 | 40.0 |
| Mean | 1.31 | 18.5 | 168 | 5.18 | 11.5 | 34.6 | 22.4 | 0.89 | 154 | 9.32 | 20.1 |
| SD | 0.49 | 8.41 | 75.3 | 1.04 | 5.48 | 18.7 | 15.8 | 0.48 | 68.3 | 4.00 | 6.67 |
| | **Mg** | **Mn** | **Mo** | **Ni** | **P** | **Pb** | **Sb** | **Se** | **Zn** | **Zr** | |
| | | | | | Metal concentrations values | | | | | | |
| | | | | | First year 2018 ($n$ = 48) | | | | | | |
| Min | 685 | 43.7 | 0.06 | 1.89 | 2699 | 0.00 | 0.01 | 0.01 | 0.01 | 0.21 | |
| Max | 1367 | 716 | 4.12 | 32.5 | 4292 | 10.6 | 13.1 | 7.85 | 92.6 | 1.19 | |
| Mean | 1030 | 298 | 1.36 | 12.5 | 3118 | 4.06 | 6.42 | 1.96 | 20.9 | 0.70 | |
| SD | 181 | 170 | 1.27 | 8.46 | 405 | 3.71 | 3.80 | 2.63 | 23.7 | 0.29 | |
| | | | | | Second year 2019 ($n$ = 48) | | | | | | |
| Min | 708 | 48.3 | 0.00 | 1.44 | 2695 | 0.00 | 0.00 | 0.00 | 0.00 | 0.23 | |
| Max | 1325 | 787 | 5.92 | 34.2 | 4393 | 11.75 | 12.9 | 8.16 | 88.6 | 1.22 | |
| Mean | 1022 | 316 | 1.25 | 12.3 | 3183 | 4.10 | 6.54 | 1.98 | 20.8 | 0.72 | |
| SD | 160 | 188 | 1.46 | 9.00 | 404 | 3.79 | 3.42 | 2.69 | 23.2 | 0.31 | |
| | | | | | Data across two years ($n$ = 96) | | | | | | |
| Min | 685 | 43.7 | 0.00 | 1.44 | 2695 | 0.00 | 0.00 | 0.00 | 0.00 | 0.21 | |
| Max | 1367 | 787 | 5.92 | 34.2 | 4393 | 11.8 | 13.1 | 8.16 | 92.55 | 1.22 | |
| Mean | 1026 | 307 | 1.30 | 12.4 | 3151 | 4.08 | 6.48 | 1.97 | 20.85 | 0.71 | |
| SD | 168 | 177 | 1.35 | 8.59 | 399 | 3.69 | 3.56 | 2.62 | 23.07 | 0.29 | |

## 3.2. Environment Pollution Indices

### 3.2.1. Contamination Factor (Cf)

Table 3 showed the descriptive statistical data of the Cf of sediment samples. According to the Cf average results, Qaroun Lake sediment samples classified as low Cf (Cf < 1) for Al, Ba, Cr, Fe, Mg, Mn, Pb, Co, Cu, Ni, Mo, and Zn. However, the Cf of As, Cd, Sb, Se, Ga, and Zr demonstrated very high contamination (6 < Cf) in the studied samples. The categories of moderate and considerable contamination were recorded for Li, P, and Hf.

**Table 3.** Statistical description of contamination factors (Cf) in Qaroun Lake sediment over two years.

| | **Al** | **As** | **Ba** | **Cd** | **Co** | **Cr** | **Cu** | **Fe** | **Ga** | **Hf** | **Li** |
|---|---|---|---|---|---|---|---|---|---|---|---|
| | | | | | **Contamination Factor Values** | | | | | | |
| | | | | | First year 2018 (*n* = 48) | | | | | | |
| Min | 0.04 | 1.87 | 0.11 | 33.0 | 0.12 | 0.07 | 0.12 | 0.06 | 1.56 | 1.31 | 0.52 |
| Max | 0.27 | 18.6 | 0.59 | 79.7 | 1.15 | 0.87 | 2.76 | 0.52 | 16.1 | 6.67 | 2.00 |
| Mean | 0.16 | 10.2 | 0.33 | 52.2 | 0.68 | 0.41 | 0.90 | 0.26 | 8.47 | 3.03 | 1.01 |
| SD | 0.06 | 4.56 | 0.15 | 11.2 | 0.31 | 0.22 | 0.64 | 0.14 | 3.76 | 1.37 | 0.34 |
| | | | | | Second year 2019 (*n* = 48) | | | | | | |
| Min | 0.04 | 2.55 | 0.09 | 34.0 | 0.05 | 0.08 | 0.14 | 0.06 | 2.02 | 1.24 | 0.65 |
| Max | 0.26 | 20.3 | 0.57 | 75.3 | 1.33 | 0.85 | 2.73 | 0.54 | 18.0 | 6.14 | 1.91 |
| Mean | 0.16 | 10.5 | 0.34 | 53.6 | 0.68 | 0.41 | 0.90 | 0.25 | 8.66 | 3.18 | 1.01 |
| SD | 0.06 | 4.93 | 0.16 | 10.4 | 0.34 | 0.22 | 0.64 | 0.14 | 3.95 | 1.34 | 0.34 |
| | | | | | Data across two years (*n* = 96) | | | | | | |
| Min | 0.04 | 1.87 | 0.09 | 33.0 | 0.05 | 0.07 | 0.12 | 0.06 | 1.56 | 1.24 | 0.52 |
| Max | 0.27 | 20.3 | 0.59 | 79.7 | 1.33 | 0.87 | 2.76 | 0.54 | 18.0 | 6.67 | 2.00 |
| Mean | 0.16 | 10.3 | 0.33 | 52.9 | 0.68 | 0.41 | 0.90 | 0.25 | 8.56 | 3.11 | 1.01 |
| SD | 0.06 | 4.67 | 0.15 | 10.7 | 0.32 | 0.22 | 0.63 | 0.14 | 3.79 | 1.33 | 0.33 |

| | **Mg** | **Mn** | **Mo** | **Ni** | **P** | **Pb** | **Sb** | **Se** | **Zn** | **Zr** |
|---|---|---|---|---|---|---|---|---|---|---|
| | | | | | **Contamination Factor values** | | | | | |
| | | | | | First year 2018 (*n* = 48) | | | | | |
| Min | 0.03 | 0.04 | 0.04 | 0.04 | 2.45 | 0.00 | 0.05 | 0.20 | 0.00 | 12.5 |
| Max | 0.06 | 0.72 | 2.75 | 0.65 | 3.90 | 0.66 | 65.4 | 157 | 1.30 | 72.2 |
| Mean | 0.04 | 0.30 | 0.90 | 0.25 | 2.84 | 0.25 | 32.1 | 39.1 | 0.29 | 42.6 |
| SD | 0.01 | 0.17 | 0.85 | 0.17 | 0.37 | 0.23 | 19.0 | 52.6 | 0.33 | 17.5 |
| | | | | | Second year 2019 (*n* = 48) | | | | | |
| Min | 0.03 | 0.05 | 0.00 | 0.03 | 2.45 | 0.00 | 0.00 | 0.00 | 0.00 | 13.8 |
| Max | 0.06 | 0.79 | 3.95 | 0.68 | 3.99 | 0.73 | 64.3 | 163 | 1.25 | 73.8 |
| Mean | 0.05 | 0.32 | 0.84 | 0.25 | 2.89 | 0.26 | 32.7 | 39.5 | 0.29 | 43.5 |
| SD | 0.01 | 0.19 | 0.97 | 0.18 | 0.37 | 0.24 | 17.1 | 53.8 | 0.33 | 19.0 |
| | | | | | Data across two years (*n* = 96) | | | | | |
| Min | 0.03 | 0.04 | 0.00 | 0.03 | 2.45 | 0.00 | 0.00 | 0.00 | 0.00 | 12.5 |
| Max | 0.06 | 0.79 | 3.95 | 0.68 | 3.99 | 0.73 | 65.4 | 163 | 1.30 | 73.8 |
| Mean | 0.04 | 0.31 | 0.87 | 0.25 | 2.86 | 0.25 | 32.4 | 39.3 | 0.29 | 43.0 |
| SD | 0.01 | 0.18 | 0.90 | 0.17 | 0.36 | 0.23 | 17.8 | 52.3 | 0.32 | 18.0 |

3.2.2. Enrichment Factor (EF)

The *EF* descriptive statistical data of sediment samples were listed in Table 4. Generally, the mainstream metals in sediment samples demonstrated a wide range of enrichment from no enrichment to severe and extremely severe enrichment. Obtained sediments from the eastern side of Qaroun Lake were extremely severely enriched in Cd, As, Ga, Zr, Sb, Se, and P according to Hanif et al.'s [75] classification of EF. Metals such as Ba, Co, Cr, Fe, Mn, Pb, Zn, and Cu ranged from minor enrichment to moderately severe enrichment in the sediment under investigation. The Qaroun Lake sediment samples were not enriched in Mg and they ranged from moderately severe enrichment to very severe enrichment in Li, Mo, Ni, and Hf.



**Table 4.** Statistical description of Enrichment Factor (EF) in Qaroun Lake sediment over two years.

| | **Al** | **As** | **Ba** | **Cd** | **Co** | **Cr** | **Cu** | **Fe** | **Ga** | **Hf** | **Li** |
|---|---|---|---|---|---|---|---|---|---|---|---|
| | | | | | **Enrichment Factor Values in Sediment** | | | | | | |
| | | | | | First year 2018 (*n* = 48) | | | | | | |
| Min | 1.00 | 8.91 | 0.90 | 239 | 1.80 | 0.87 | 1.46 | 0.73 | 18.9 | 14.0 | 4.06 |
| Max | 1.00 | 349 | 3.50 | 779 | 8.61 | 4.41 | 12.4 | 3.01 | 110 | 35.9 | 17.8 |
| Mean | 1.00 | 86.9 | 2.14 | 362 | 4.24 | 2.45 | 5.24 | 1.54 | 55.7 | 19.2 | 6.99 |
| SD | 0.00 | 84.2 | 0.71 | 139 | 1.87 | 1.00 | 2.54 | 0.57 | 27.0 | 6.00 | 3.42 |
| | | | | | Second year 2019 (*n* = 48) | | | | | | |
| Min | 1.00 | 11.8 | 0.81 | 231 | 0.58 | 0.96 | 1.79 | 0.76 | 25.1 | 13.7 | 4.02 |
| Max | 1.00 | 394 | 3.27 | 874 | 7.94 | 4.56 | 11.8 | 2.90 | 110 | 32.0 | 18.2 |
| Mean | 1.00 | 92.7 | 2.15 | 379 | 4.22 | 2.48 | 5.32 | 1.56 | 57.2 | 20.2 | 7.01 |
| SD | 0.00 | 94.6 | 0.74 | 158 | 1.87 | 0.97 | 2.47 | 0.53 | 26.7 | 4.72 | 3.36 |
| | | | | | Data across two years (*n* = 96) | | | | | | |
| Min | 1.00 | 8.91 | 0.81 | 231 | 0.58 | 0.87 | 1.46 | 0.73 | 18.9 | 13.7 | 4.02 |
| Max | 1.00 | 394 | 3.50 | 874 | 8.61 | 4.56 | 12.4 | 3.01 | 110 | 35.9 | 18.2 |
| Mean | 1.00 | 89.8 | 2.14 | 370 | 4.23 | 2.46 | 5.28 | 1.55 | 56.5 | 19.7 | 7.00 |
| SD | 0.00 | 88.2 | 0.71 | 147 | 1.84 | 0.97 | 2.46 | 0.54 | 26.4 | 5.33 | 3.34 |

| | **Mg** | **Mn** | **Mo** | **Ni** | **P** | **Pb** | **Sb** | **Se** | **Zn** | **Zr** |
|---|---|---|---|---|---|---|---|---|---|---|
| | | | | | **Enrichment Factor values in sediment** | | | | | |
| | | | | | First year 2018 (*n* = 48) | | | | | |
| Min | 0.17 | 0.88 | 0.47 | 0.31 | 10.4 | 0.00 | 0.61 | 0.89 | 0.00 | 134 |
| Max | 0.81 | 5.65 | 21.0 | 2.91 | 69.0 | 3.59 | 531 | 1948 | 5.84 | 451 |
| Mean | 0.32 | 3.11 | 6.43 | 1.50 | 21.6 | 1.36 | 203 | 419 | 1.71 | 272 |
| SD | 0.17 | 1.51 | 6.83 | 0.79 | 14.4 | 1.22 | 125 | 668 | 1.67 | 81.3 |
| | | | | | Second year 2019 (*n* = 48) | | | | | |
| Min | 0.18 | 1.00 | 0.00 | 4.02 | 10.0 | 0.00 | 0.00 | 0.00 | 0.00 | 121 |
| Max | 0.96 | 6.37 | 26.4 | 18.2 | 73.4 | 3.97 | 519 | 2032 | 5.40 | 452 |
| Mean | 0.34 | 3.33 | 6.34 | 7.01 | 22.3 | 1.36 | 214 | 435 | 1.70 | 281 |
| SD | 0.20 | 1.68 | 7.32 | 3.36 | 15.5 | 1.24 | 121 | 689 | 1.66 | 86.4 |
| | | | | | Data across two years (*n* = 96) | | | | | |
| Min | 0.17 | 0.88 | 0.00 | 0.31 | 10.0 | 0.00 | 0.00 | 0.00 | 0.00 | 121 |
| Max | 0.96 | 6.37 | 26.4 | 18.2 | 73.4 | 3.97 | 531 | 2032 | 5.84 | 452 |
| Mean | 0.33 | 3.22 | 6.38 | 4.25 | 22.1 | 1.36 | 209 | 427 | 1.71 | 277 |
| SD | 0.18 | 1.57 | 6.96 | 3.69 | 14.7 | 1.21 | 121 | 668 | 1.64 | 82.6 |

### 3.2.3. Geoaccumulation Index ($I_{geo}$)

Table 5 shows the descriptive statistical data of $I_{geo}$ in studied sediment samples. Müller [76,77] classified the sediment into seven categories (Table 1) according to the geoaccumulation index values from unpolluted to extremely polluted. The studied Qaroun Lake sediment samples were unpolluted by Al, Ba, Co, Cr, Fe, Mg, Mn, Ni, Pb, and Zn. However, sediment samples ranged from strongly polluted to extremely polluted by Cd, Zr, Sb, and Se. Moreover, samples varied from unpolluted to moderately polluted by Hf, Mo, Li, and P. For As and Ga in the examined sediment samples varied between moderately polluted to strongly polluted.

**Table 5.** Statistical description of Geoaccumulation index ($I_{geo}$) in Qaroun Lake sediment over two years.

| | **Al** | **As** | **Ba** | **Cd** | **Co** | **Cr** | **Cu** | **Fe** | **Ga** | **Hf** | **Li** |
|---|---|---|---|---|---|---|---|---|---|---|---|
| | **Geoaccumulation Index Values in Sediment** | | | | | | | | | | |
| | First year 2018 (*n* = 48) | | | | | | | | | | |
| Min | −5.15 | 0.32 | −3.76 | 4.46 | −3.64 | −4.39 | −3.64 | −4.54 | 0.06 | −0.20 | −1.54 |
| Max | −2.47 | 3.63 | −1.35 | 5.73 | −0.38 | −0.78 | 0.88 | −1.53 | 3.42 | 2.15 | 0.42 |
| Mean | −3.33 | 2.56 | −2.32 | 5.09 | −1.37 | −2.16 | −1.10 | −2.79 | 2.30 | 0.88 | −0.64 |
| SD | 0.67 | 0.87 | 0.74 | 0.30 | 0.95 | 1.01 | 1.12 | 0.91 | 0.87 | 0.63 | 0.45 |
| | Second year 2019 (*n* = 48) | | | | | | | | | | |
| Min | −5.27 | 0.77 | −4.04 | 4.50 | −5.01 | −4.28 | −3.38 | −4.62 | 0.43 | −0.27 | −1.21 |
| Max | −2.50 | 3.76 | −1.39 | 5.65 | −0.18 | −0.82 | 0.87 | −1.47 | 3.58 | 2.03 | 0.35 |
| Mean | −3.34 | 2.61 | −2.34 | 5.13 | −1.46 | −2.15 | −1.08 | −2.78 | 2.35 | 0.96 | −0.64 |
| SD | 0.71 | 0.82 | 0.79 | 0.29 | 1.20 | 1.00 | 1.08 | 0.91 | 0.81 | 0.64 | 0.45 |
| | Data across two years (*n* = 96) | | | | | | | | | | |
| Min | −5.27 | 0.32 | −4.04 | 4.46 | −5.01 | −4.39 | −3.64 | −4.62 | 0.06 | −0.27 | −1.54 |
| Max | −2.47 | 3.76 | −1.35 | 5.73 | −0.18 | −0.78 | 0.88 | −1.47 | 3.58 | 2.15 | 0.42 |
| Mean | −3.34 | 2.59 | −2.33 | 5.11 | −1.41 | −2.15 | −1.09 | −2.79 | 2.33 | 0.92 | −0.64 |
| SD | 0.68 | 0.84 | 0.75 | 0.29 | 1.07 | 0.99 | 1.08 | 0.89 | 0.83 | 0.62 | 0.44 |

| | **Mg** | **Mn** | **Mo** | **Ni** | **P** | **Pb** | **Sb** | **Se** | **Zn** | **Zr** |
|---|---|---|---|---|---|---|---|---|---|---|
| | **Geoaccumulation index values in sediment** | | | | | | | | | |
| | First year 2018 (*n* = 48) | | | | | | | | | |
| Min | −5.65 | −5.10 | −5.23 | −5.31 | 0.71 | −11.2 | −4.91 | −2.91 | −13.4 | 3.06 |
| Max | −4.66 | −1.07 | 0.87 | −1.21 | 1.38 | −1.18 | 5.45 | 6.71 | −0.20 | 5.59 |
| Mean | −5.09 | −2.60 | −1.57 | −3.00 | 0.91 | −4.24 | 3.72 | 2.02 | −4.26 | 4.70 |
| SD | 0.26 | 1.00 | 1.88 | 1.24 | 0.17 | 3.66 | 2.46 | 3.72 | 3.94 | 0.67 |
| | Second year 2019 (*n* = 48) | | | | | | | | | |
| Min | −5.61 | −4.96 | −4.81 | −5.70 | 0.71 | −6.32 | 2.55 | −0.10 | −4.18 | 3.06 |
| Max | −4.70 | −0.93 | 1.40 | −1.13 | 1.41 | −1.03 | 5.42 | 6.77 | −0.26 | 5.59 |
| Mean | −5.09 | −2.53 | −1.16 | −3.10 | 0.94 | −2.55 | 4.38 | 4.52 | −2.51 | 4.70 |
| SD | 0.23 | 1.02 | 1.59 | 1.39 | 0.17 | 1.41 | 0.75 | 2.10 | 1.17 | 0.67 |
| | Data across two years (*n* = 96) | | | | | | | | | |
| Min | −5.65 | −5.10 | −5.23 | −5.70 | 0.71 | −11.23 | −4.91 | −2.91 | −13.38 | 3.06 |
| Max | −4.66 | −0.93 | 1.40 | −1.13 | 1.41 | −1.03 | 5.45 | 6.77 | −0.20 | 5.59 |
| Mean | −5.09 | −2.57 | −1.38 | −3.05 | 0.92 | −3.49 | 4.04 | 2.98 | −3.48 | 4.70 |
| SD | 0.24 | 0.99 | 1.74 | 1.30 | 0.17 | 2.96 | 1.85 | 3.38 | 3.11 | 0.66 |

### 3.2.4. Degree of Contamination (Dc), Pollution Load Index (PLI), and Potential Ecological Risk Index (RI)

The Dc, PLI, and RI descriptive statistical results of sediment samples are presented in Table 6. All tested sediment samples were very highly contaminated by metals agreeing with the data of (Dc) in Table 7. While the results of PLI showed that about 59% of the Qaroun Lake sediment samples were polluted (PLI > 1), 41% of samples were unpolluted (PLI < 1), as shown in Table 7. Finally, the results of the potential ecological risk index showed that all analyzed samples were classified as very high ecological risk (RI > 600) in Table 7. Figures 2 and 3 show the spatial distribution of Dc, PLI, and RI of tested metals in Qaroun Lake sediment samples in years 2018 and 2019. The samples of the eastern and middle parts of the lake and near the mouths of drains proved extremely polluted.

**Table 6.** Statistical description of the degree of contamination (Dc), pollution load index (PLI), and potential ecological risk index (RI) in Qaroun Lake sediment over two years.

|  | Dc | PLI | RI |
|---|---|---|---|
| **First year 2018 (*n* = 48)** | | | |
| Min | 149 | 0.20 | 1139 |
| Max | 297 | 1.60 | 2478 |
| Mean | 196 | 0.97 | 1679 |
| SD | 44.0 | 0.37 | 330 |
| **Second year 2019 (*n* = 48)** | | | |
| Min | 148 | 0.35 | 1176 |
| Max | 309 | 1.82 | 2353 |
| Mean | 200 | 1.07 | 1725 |
| SD | 44.0 | 0.39 | 303 |
| **Data across two years (*n* = 96)** | | | |
| Min | 148 | 0.20 | 1139 |
| Max | 309 | 1.82 | 2478 |
| Mean | 198 | 1.02 | 1702 |
| SD | 43.3 | 0.38 | 313 |

**Table 7.** Assessment of surface sediments of Qaroun Lake according to categories of degree of contamination (Dc), pollution load index (PLI), and potential ecological risk index (RI).

| Indices | Classes | Sediment Samples (%) | | |
|---|---|---|---|---|
| | | 1st Year (2018) | 2nd Year (2019) | Across Two Years |
| Degree of Contamination (Dc) | Low | 0 | 0 | 0 |
| | Moderate Dc | 0 | 0 | 0 |
| | Considerable Dc | 0 | 0 | 0 |
| | Very high Dc | 100% (48 samples) | 100% (48 samples) | 100% (96 samples) |
| Pollution Load Index (PLI) | Unpolluted | 37.5% (18 samples) | 43.75% (21 samples) | 40.63% (39 samples) |
| | Polluted | 62.5% (30 samples) | 56.25% (27 samples) | 59.38% (57 samples) |
| Ecological Risk Index (RI) | Low ecological risk | 0 | 0 | 0 |
| | Moderate ecological risk | 0 | 0 | 0 |
| | Considerable ecological risk | 0 | 0 | 0 |
| | Very high ecological risk | 100% (48 samples) | 100% (48 samples) | 100% (96 samples) |

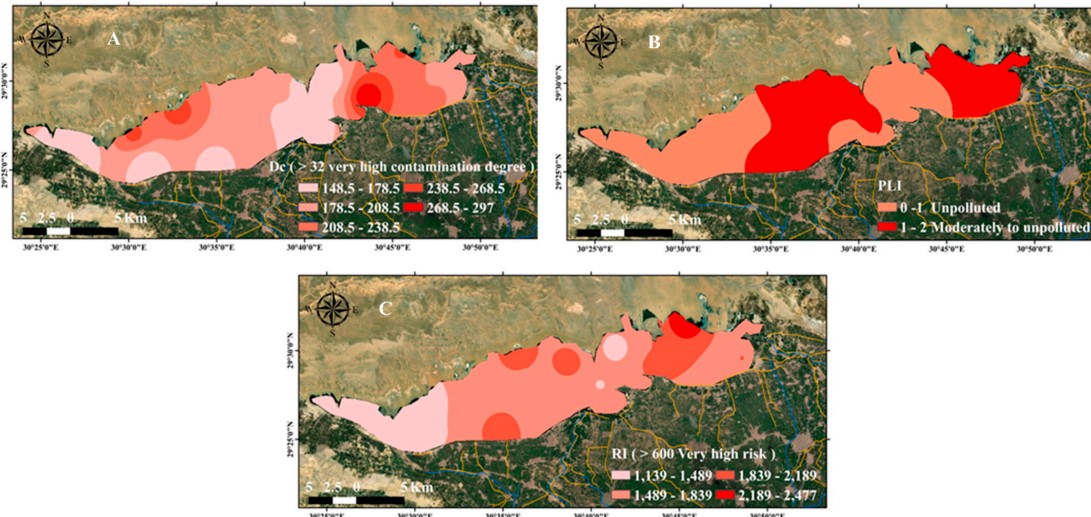

**Figure 2.** Spatial distribution maps of sediment quality indices: (**A**). Degree of contamination (Dc); (**B**). Pollution load index (PLI); and (**C**). potential risk index (RI) in year 2018.

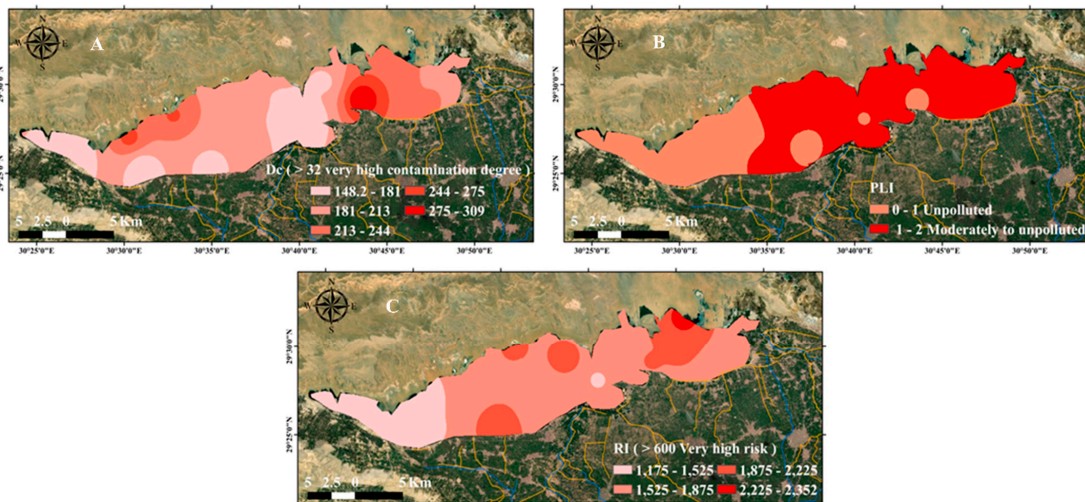

**Figure 3.** Spatial distribution maps of sediment quality indices: (**A**) Degree of contamination (Dc); (**B**) Pollution load index (PLI); and (**C**) potential risk index (RI) in the year 2019.

### 3.3. Correlation Matrix

Pearson correlation coefficient analysis shown in Figure 4 showed a strong positive correlation of Cd, Cr, Cu, Fe, Pb, Ni, Co, Li, Hf, Zr, Ba, and Mn with Al (r = 0.76, 0.73, 0.71, 0.8, 0.54, 0.75, 0.61, 0.65, 0.83, 0.64, 0.64, and 0.51 respectively) and Sb, Cd, Cr, Cu, Pb, Zn, Ni, Co, Li, and Hf with Fe (r = 0.63, 0.89, 0.84, 0.92, 0.60, 0.82, 0.83, 0.81, 0.67, 0.89 respectively); as well as Cr, Co, Ba, Zr, and Ga with Mn (r = 0.63, 0.53, 0.67, 0.74, 0.53 respectively) and Sb, Cd, Cr, Cu, Pb, Zn, Ni, Co, and Zr with Hf (r = 0.51, 0.89, 0.91, 0.93, 0.65, 0.77, 0.89, 0.79, 0.68 respectively). Another significant relation of Zn with Sb (r = 0.66), Cd (r = 0.74), Cr (r = 0.71), Cu (r = 0.92), Pb (r = 0.71), Ni (r = 0.81), Co (r = 0.66), and Li (r = 0.54) was stated. However, As and Se showed a negative relation with Al (r = −0.61 and −0.58), Fe (r = −0.39 and −0.71), Mn (r = −0.38 and −0.62), Hf (r = −0.58 and −0.64), Zn (r = −0.18 and −0.52), Cu (r = −0.51 and −0.67), Ni (r = −0.62 and −0.71), Ba (r = −0.66 and −0.34), Zr (r = −0.69 and −0.35), Cr (r = −0.49 and −0.77), Pb (r = −0.33 and −0.48), Sb (r = −0.07 and −0.51), and Co (r = −0.49 and −0.78).

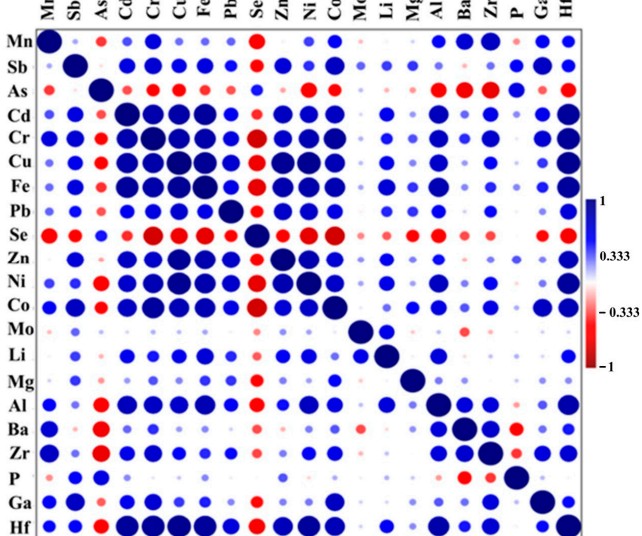

**Figure 4.** Pearson correlation matrix of heavy metals in surface sediments. For colour, blue colour means positive correlation and red colour means negative correlation. The size of the circle refers to the degree of correlation.

### 3.4. Multivariate Statistical Analysis

3.4.1. Cluster Analysis

Figure 5 shows the dendrogram resulting from the cluster analysis of metal concentrations in Qaroun Lake sediments. There are four main groups of clustering that have been detected in the CA results for tested elements. The first one is a grouping of the metals Al, Fe, and Zr (Cluster I), and the second one is composed of P (Cluster II). The third cluster (Cluster III) included Mg. The last group (Cluster IV) was further split into two sub-clusters, one group containing Mn and the other one grouping Sb, Cd, Pb, Se, Mo Ni, Co, Hf, As, Li Cr, Cu, Zn, Ba, and Ga.

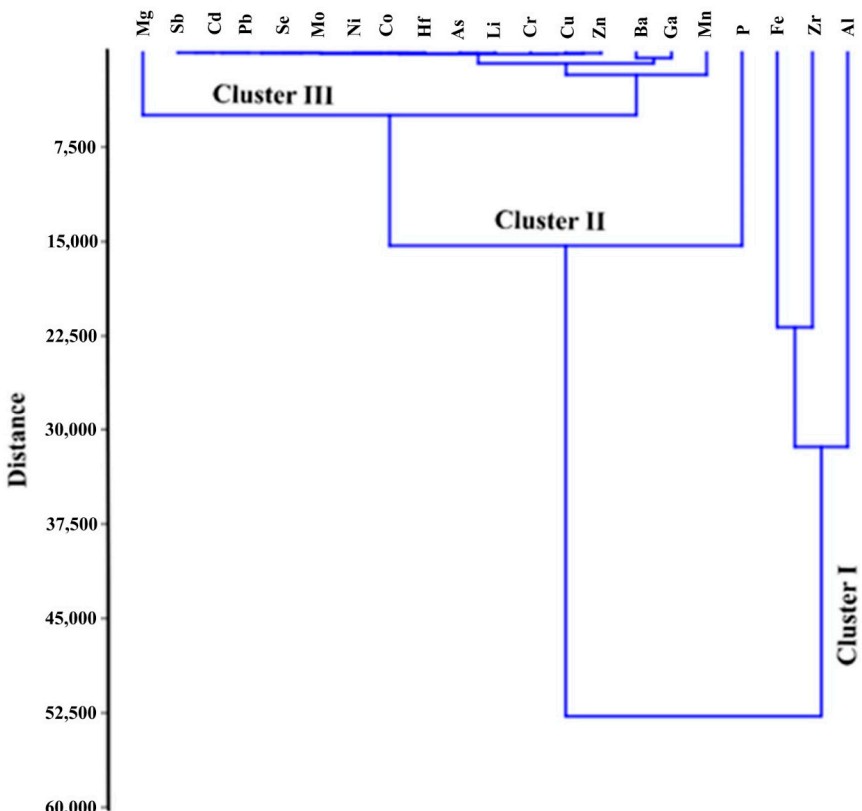

**Figure 5.** Dendrogram showing the clusters of metals in Qaroun Lake sediment samples.

3.4.2. Principal Component Analysis

The results of a PCA for the 21 tested elements in Qaroun Lake sediment samples are shown in Table 8 and Figure 6. Three significant components explain in total 99.776% of the variance were selected. The first component (PC1) explained 80.998% of the total variance was predominated by large positive loading of Al, Fe, and Zr (loading 0.69767, 0.66589, and 0.26369 respectively) especially from sampling sites S1, S2, S3, S6, S7, S8, S14, and S16. The second principal component (PC2) explained 13.696% of the total variance and mainly composed of P, Mn, Ba, Mg, Zn, and Ga (loading −0.09366, 0.032219, 0.016643, −0.012688, −0.003363, and 0.0055782 respectively). The third (PC3) one explained 5.0824% of the total variance.

**Table 8.** Principal component analysis results.

| Variable | PC1 | PC2 | PC3 |
|---|---|---|---|
| Mn | 0.012694 | 0.032219 | 0.019371 |
| Sb | 0.000304 | $-0.00042$ | 0.001246 |
| As | $-0.0007$ | $-0.00129$ | $-0.00061$ |
| Cd | 0.000138 | $-4.19 \times 10^{-5}$ | 0.000193 |
| Cr | 0.002398 | $8.72 \times 10^{-5}$ | 0.004486 |
| Cu | 0.001997 | $-0.00121$ | 0.002912 |
| Fe | 0.66589 | $-0.61233$ | 0.41661 |
| Pb | 0.000316 | $7.51 \times 10^{-5}$ | 0.000377 |
| Se | $-0.00028$ | $9.63 \times 10^{-5}$ | $-0.0002$ |
| Zn | 0.002336 | $-0.00336$ | 0.005295 |
| Ni | 0.001044 | $-0.00029$ | 0.00068 |
| Co | 0.000573 | $-0.00023$ | 0.001358 |
| Mo | $2.23 \times 10^{-5}$ | $1.08 \times 10^{-6}$ | $-0.00012$ |
| Li | 0.000608 | $-0.00063$ | $-0.001$ |
| Mg | 0.009468 | $-0.01269$ | 0.013791 |
| Al | 0.69767 | 0.31179 | $-0.64028$ |
| Ba | 0.00606 | 0.016643 | $-0.00567$ |
| Zr | 0.26369 | 0.71941 | 0.64055 |
| P | $-0.00122$ | $-0.09366$ | 0.070073 |
| Ga | 0.003915 | 0.005578 | 0.024705 |
| Hf | 0.00054 | $8.70 \times 10^{-5}$ | 0.000666 |
| Eigenvalue | $4.57 \times 10^{7}$ | $7.73 \times 10^{6}$ | $2.87 \times 10^{6}$ |
| % Total variance | 80.998 | 13.696 | 5.0824 |
| Cumulative % variance | 80.998 | 94.694 | 99.7764 |

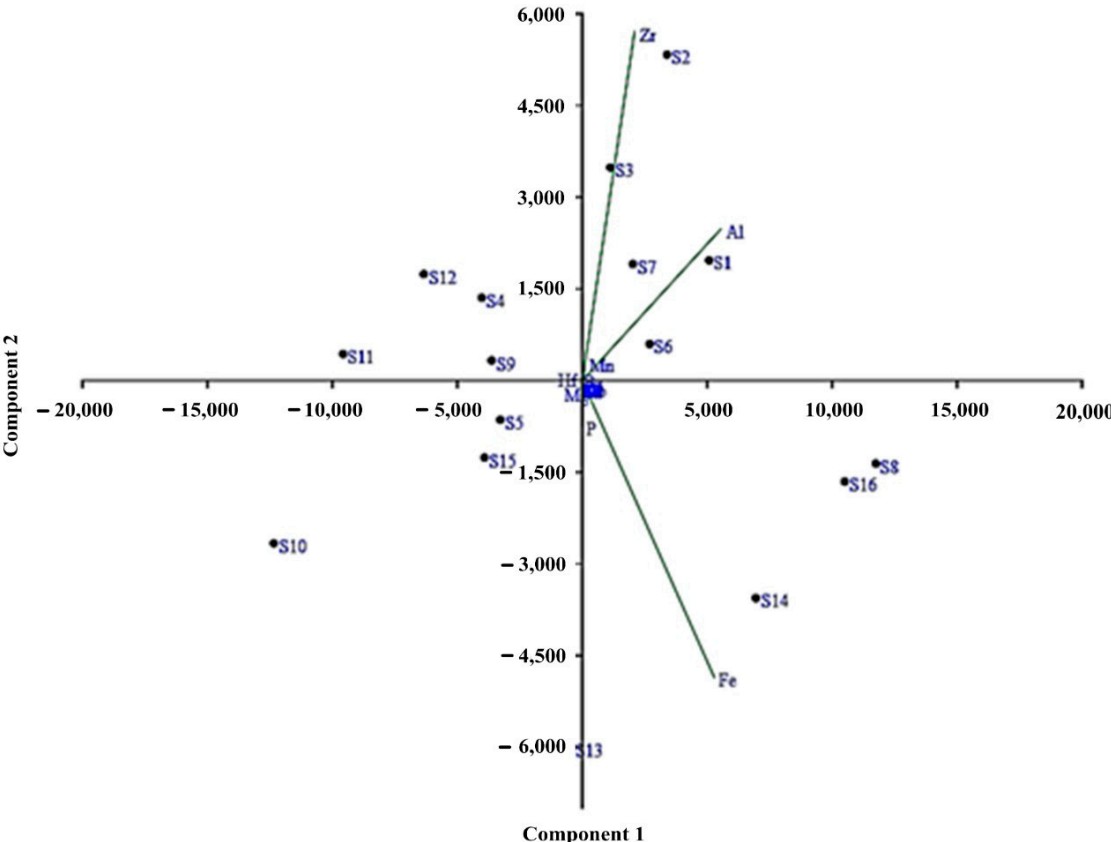

**Figure 6.** Principal component analysis for metals of Qaroun Lake sediment samples.

*3.5. Performance Efficiency of PLSR and MLR Models to Predict the Degree of Contamination (Dc), Pollution Load Index (PLI), and Potential Ecological Risk Index (RI)*

In Table 9, the calibration (Cal.) models presented the best performance efficiency to predict the PLI, RI and Dc, based on the 21 selected elements, with $R^2_{cal}$ = 0.961–0.991 and with $ACC_c$ = 0.981–0.999 for PLSR, and with $R^2_{cal}$ = 0.947–0.996 and with $ACC_c$ = 0.976–0.998 for MLR. The validation (Val.) models also presented the best performance efficiency, with $R^2_{val}$ = 0.948–0.989 and with $ACC_v$ = 0.984–0.999 for PLSR, and with $R^2_{val}$ = 0.760–0.979 and with $ACC_v$= 0.867–0.984 for MLR. The Val. Model of the PLSR method performed better efficiency to predict the PLI than the Val. Model of the MLR method (Table 9; Figures 7 and 8). In general, the Cal. and Val. models of the PLSR method showed barely higher values of $R^2$ and ACC and lower values of RMSE and MAD than the MLR method for RI and Dc. For example, the $R^2_{val}$, $ACC_v$, $RMSE_v$, and $MAD_v$ for RI of PLSR models were 0.951, 0.989, 70.270, and 55.823, respectively, while the $R^2_{val}$, $ACC_v$, $RMSE_v$, and $MAD_v$ for RI of MLR models were 0.930, 0.955, 117.487, and 93.095, respectively. The optimum LVs were 3, 5, and 3 for the calibration PLSR models of PLI, RI, and Dc, respectively. A comparison between measuring datasets, calibrating datasets, and validating datasets for three pollution indices using the PLSR and MLR models is shown in Figures 7 and 8. Hence, the results showed that there were no clear overfitting and underfitting between measuring, calibrating, and validating datasets for PLI, RI and Dc. Moreover, the models of Cal and Val of both PLSR and MLR showed good slopes between measured and predicted data of the PLI, RI, and Dc and the value of the slope of the two models varied from 0.864 to 1.087. Thus, these results confirm that PLSR and MLR models including 21 elements as input data can be used as alternative methods to estimate the three pollution indices.

**Table 9.** Results of calibration ($R^2_{cal}$, $RMSE_C$, $MAD_c$, and $Acc_c$), and ten-fold cross-validation ($R^2_{val}$, $RMSE_v$, $MAD_v$, and $Acc_v$): PLSR and multiple linear regression MLR models of the relationships between several heavy metals and three pollution load indices (PLI, RI and Dc). ***: $p < 0.001$.

| Models | Pollution Indices | LVs | Calibration | | | | Validation | | | |
|---|---|---|---|---|---|---|---|---|---|---|
| | | | $R^2_{cal}$ | $RMSE_c$ | $MAD_c$ | $ACC_c$ | $R^2_{val}$ | $RMSE_v$ | $MAD_v$ | $ACC_v$ |
| PLSR | Dc | 3 | 0.991 *** | 4.0165 | 3.1635 | 0.999 | 0.989 *** | 4.367 | 3.504 | 0.999 |
| | PLI | 3 | 0.961 *** | 0.084 | 0.060 | 0.981 | 0.948 *** | 0.099 | 0.076 | 0.984 |
| | RI | 5 | 0.967 *** | 59.097 | 47.025 | 0.988 | 0.951 *** | 70.270 | 55.823 | 0.989 |
| MLR | Dc | - | 0.996 *** | 3.117 | 1.469 | 0.998 | 0.979 *** | 7.575 | 5.688 | 0.984 |
| | PLI | - | 0.947 *** | 0.086 | 0.073 | 0.989 | 0.760 *** | 0.231 | 0.173 | 0.867 |
| | RI | - | 0.964 *** | 70.664 | 51.031 | 0.976 | 0.930 *** | 117.487 | 93.095 | 0.955 |

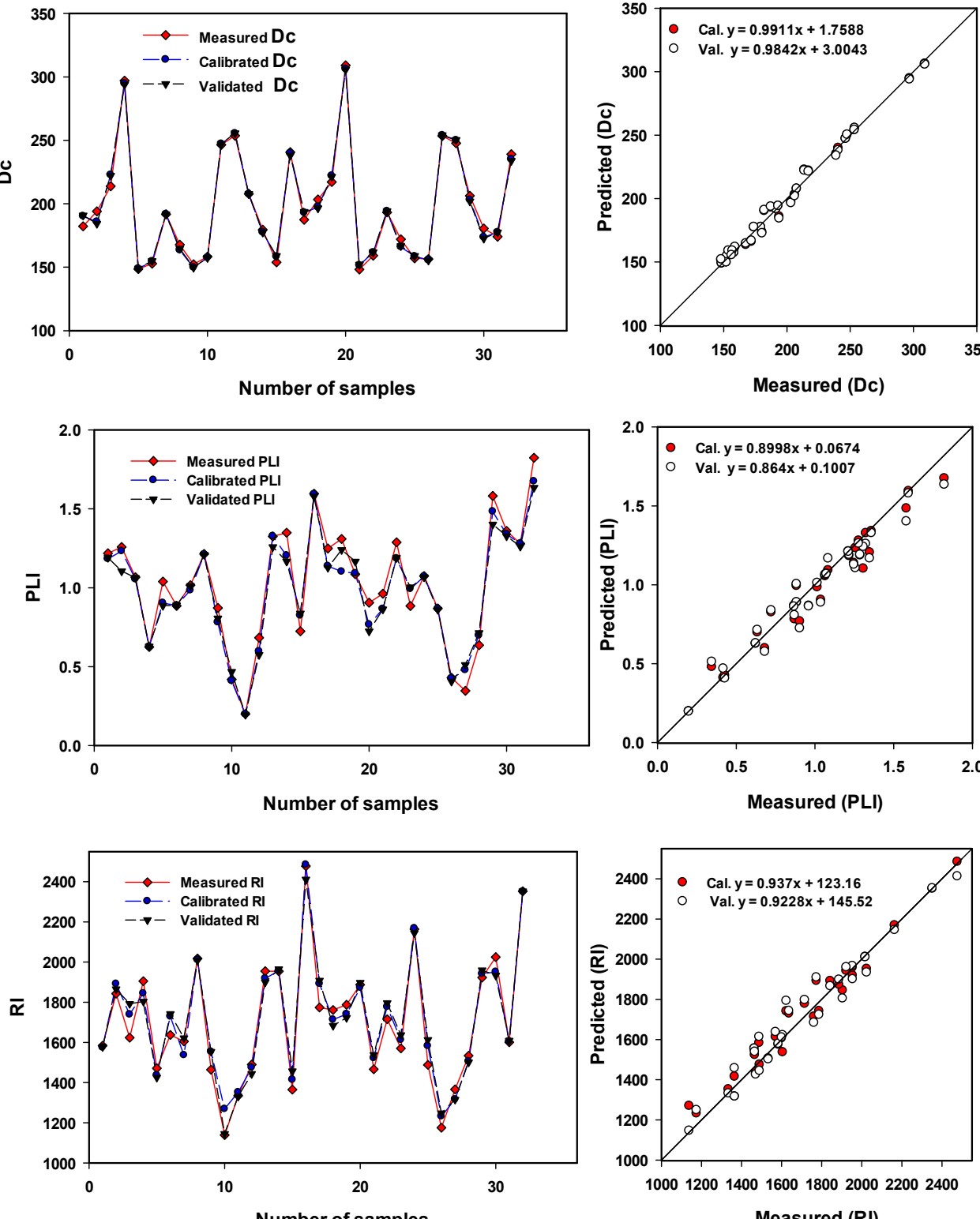

**Figure 7.** Comparison between measuring datasets, calibrating datasets, and validating datasets for three pollution indices index using the PLSR models based on several heavy metals. Statistical analysis was showed in Table 9.

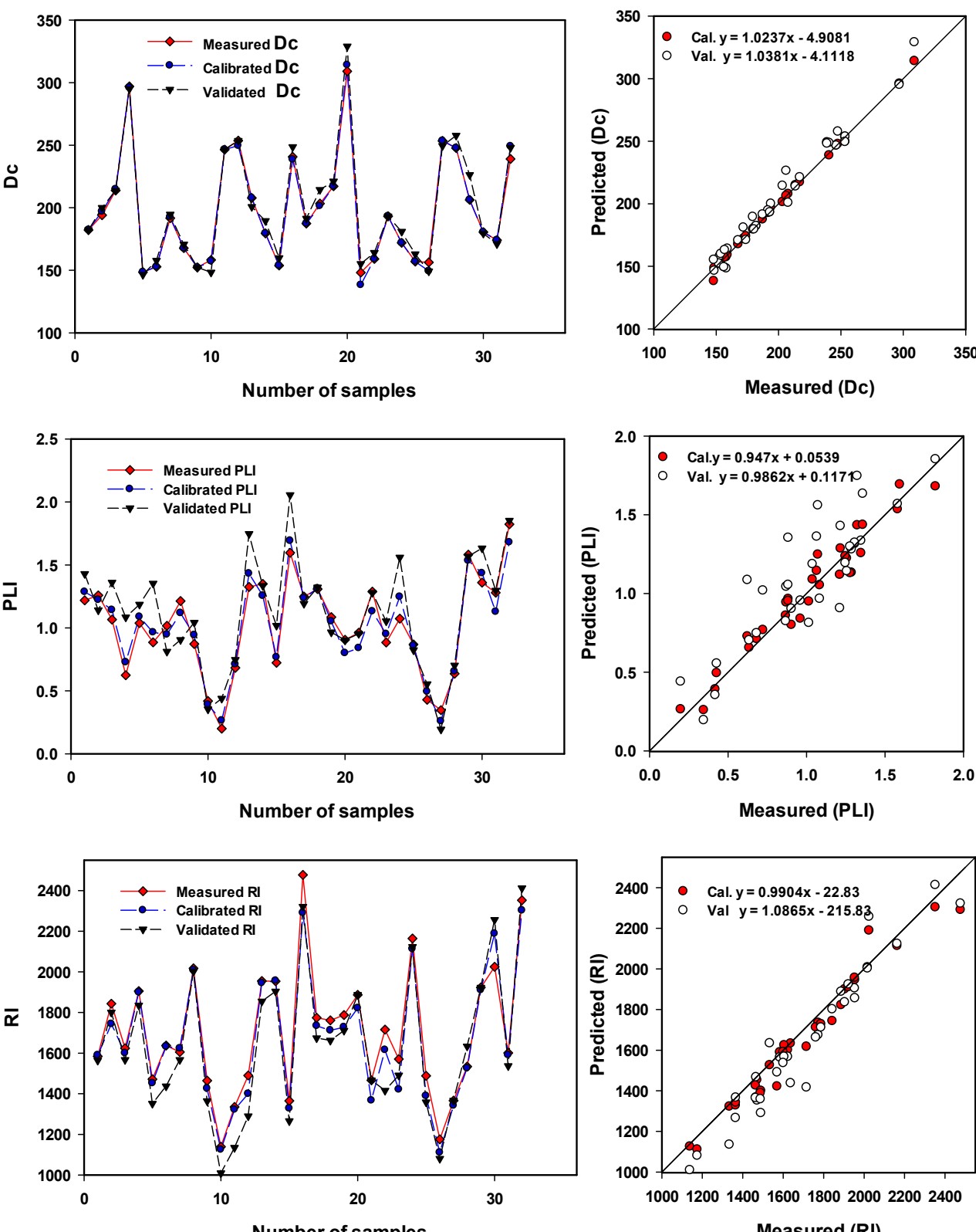

**Figure 8.** Comparison between measuring datasets, calibrating datasets, and validating datasets for three pollution indices index using the MLR models based on several heavy metals. Statistical analysis was showed in Table 9.

## 4. Discussion

The fluctuation in heavy metal concentrations could be related to changes in element sources and current physico-chemical conditions. Adsorption, precipitation, and

redox changes all have an impact on the metal content in sediments [90]. The distribution and concentrations of pollutants as the heavy metals in sediments variate with grain size, percentage of silicates, oxide-hydroxide, carbonates, and organic matter in sediments [25,91,92]. The sediments samples were collected from the eastern and middle parts of the lake (at the mouth of El-Bates and El-Wadi drains, respectively) have more organic matter than those of the western side, and the northern desert coast is far from the El-Bates drain. A similar observation was previously confirmed by Soliman et al. [24], El-Zeiny et al. [20], and El-Kady et al. [12]. The twenty-one examined element concentration results showed that all environmental sensitive metals in the lake sediments appeared to have gathered in front of the El-Bats and El-Wadi drains, indicating high levels in the lake's eastern and central regions. These locations receive a massive amount of agricultural, industrial, aquacultural, and untreated sewage drainage water [12,16,17,93]. Previously, Abdel Wahed et al. [18] documented a significant relationship between the toxic metals in the lake sediments and those in the drain water. Further, Bai et al. [94] stated that heavy metals, such as Cd, Cr, Cu, Ni, Pb, and Zn, were associated with anthropogenic inputs of domestic sewage and agrochemical discharge. Therefore, the primary source of environment-sensitive metals in the lake, especially in its eastern and northeastern sections, comes from the El-Bats drain. The vulnerability of the eastern and southern side of the lake caused by the accumulation of metals in lake sediments was previously confirmed by El-Sayed et al. [10], Attia et al. [11], and El-Kady et al. [12]. Coagulation due to the mixing of freshwater and saltwater at the inlet of drains that pours into the lake will lead to appreciable sedimentation at the mouth of the drains and increase the percentage of fine sediments and organic matter. Consequently, it may be an additional probable factor for an elevated concentration of metals [92,95]. Organic compounds and oxidation-reduction results had an impact on heavy metals dissolution and sorption [96]. Arid and semi-arid locations with high evaporation rates, such as the research area, have oxidizing aquatic ecosystems. In these areas, excessive salinity and pH are common [97,98]. Meanwhile, the locations with a lot of organic matter, such as the areas under investigation near the inlet of drains, represent a reducing aquatic ecosystem. Due to microbial reduction, reducing zones may have low dissolved heavy metals, and provide the conditions for the precipitation of heavy metals [99]. El-Zeiny et al. [20] stated a significant positive correlation between the lake's water electric conductivity and heavy metals in bottom sediment, which refers to the salinity of water impact in the accumulation of heavy metals in the bottom sediments. The salinity of lake water in the western and northwestern parts was higher than in the eastern side [18]. Therefore, the lake water salinity may be a possible factor controlling the concentration of metals in the eastern and middle parts of the lake. The high levels of Al and Fe in the sediments studied suggested spontaneous contamination by terrigenous material [9,100].

Environmental pollution indices represent an effective tool and guide for evaluating the state of the sediments environment by using a comprehensive geochemical examination [101]. The pollution indices were grouped into six classes by Kowalska et al. [48] based on the diverse aims of calculation, i.e., to offer information about: (1) contamination levels from each of the metals studied on an individual basis (CF and $I_{geo}$); (2) the total amount of pollution (Dc and PLI); (3) the heavy metal sources (EF); (4) the potential ecological risk (RI); (5) the place with the greatest potential risk of accumulating heavy metal; (6) the surface horizon's capacity to accumulate heavy metals. The indices play a beneficial role in determining the accumulation of metals caused by natural processes or anthropogenic actions [48]. The results of (Cf) revealed that the sediment samples that were thoroughly investigated have a wide range of metal contamination levels, ranging from very low to very high. In the testing lake sediments, contamination levels of As, Cd, Sb, Se, Ga, and Zr were very high. The findings of ($I_{geo}$), which demonstrated that the lake sediments were significantly polluted by the same metals mentioned previously, confirmed these conclusions. The EF value of naturally derived elements is almost less than one, whereas the EF values of anthropogenically derived elements are several orders of magnitude higher [102].

Excluding the Al normalized element, the EF results refer to all the investigated metals having anthropogenic sources, and the Mg was the only investigated element that has a natural origin. According to the Håkanson [66] criteria that were utilized to identify the contamination degree (Dc) of the lake sediments resulting from all metals presence, the results of Qaroun Lake sediments represent a very high level of contamination (Dc > 32). The (PLI) outcomes revealed that the eastern and southern sides of the middle part of the lake were polluted, and the western and west northern parts were still unpolluted. The (RI) data showed that the bottom sediment of Qaroun Lake was a very high ecological risk due to the presence of metals such as As, Cd, Cu, Ni, Co, and Pb, which are prospective environmental poisonous metals.

Pearson's correlation matrix is commonly used to determine the dimension of similarity and to assess the interrelationships between the parts being studied [103]. The correlation matrix revealed that some elements are highly associated with Al. Al is one of the most well-preserved elements because of its high resistance to weathering and erosion. These referred to these elements are mainly originated from natural sources. On the other hand, they may be of anthropogenic sources and bind to clay minerals [104–106]. Furthermore, some metals are strongly correlated with Fe and Mn. Regarding the factors controlling the distribution of the heavy metals, the results mentioned confirm that clay minerals and (oxy) hydroxides of iron and manganese play roles in the distribution of metals in examined samples. The metals investigated from the Qaroun Lake were negatively linked with As, Se, and P, indicating that these metals may have different sources and distributions.

To comprehend the origins of elements, a single correlation study is insufficient. As a result, CA and PCA have been widely utilized to detect element sources in sediment and their element properties [107]. The cluster analysis seems to be a useful technique to collect a range of data sets by grouping them. Principal component analysis refers to an interaction of various observed variables that explain the behaviour of a single process that links these variables together [108]. The results of the PCA showed a high loading of Al, Fe, and Zr in the three main components, and other tested metals in the same components had significant positive correlations with Al, Fe, and Mn, indicating these metals had a natural occurrence in sediment or from anthropogenic sources, with the distribution controlled by clay minerals and oxides of iron and Mn. Moreover, the tested metals had a positive correlation with the lake's eastern and southern parts samples.

Therefore, all the previously mentioned finding speculates that metals in Qaroun Lake sediment may originate from a mixed source (natural and anthropogenic). The findings suggest that the potential environmental toxic elements As, Se, Cd, Zr, Hf, Ga, Sb, Cr, Ni, and P in the bottom sediment samples of Qaroun Lake mainly originated from anthropogenic activity. Human activities, such as industrial, agricultural, entertainment, aquacultural, traffic, and urban, surrounding Qaroun Lake are essential and supposed to be potential sources for the enrichment of toxic metals in lake sediments. The ceramic and fertilizer industries may be expected sources for harmful elements, such as Zr, Hf, and Ga. The anomalous concentrations of Zr and Hf may be due to natural resources, such as heavy minerals in the bottom sediments. The heavy mineral grains may derive from Western Desert sand dunes grain settling in the lake. The sand dunes composed more than 1% of heavy minerals [109]. Continuing to get rid of these emissions without treatment leads to an increase in the proportion of environmentally harmful metals in the lake, leads to the destruction of the different ecological systems in the lake, and will cause a complete cessation of fishing and recreational activities, as well as a direct impact on migratory birds. BirdLife International added Lake Qarun to its list of Important Bird Areas (IBA) in 1999, recognizing its international significance for bird conservation during their migration road in the east of Africa and the Middle east. In the winter, Qarun Lake attracts massive numbers of waterfowl, with 32,665 recorded in 1989/90. It is known to breed at least ten species of waterbirds that began nesting at Qarun Lake in the early 1990s, and an estimated 1000 couples nested on El Qarn island in summer

1998 [110,111]. So, it is urgent to establish integrated treatment plants (tertiary treatment) to treat the wastewater before discharging into the lake and prioritize the implementation of environmental regulations relating to the treatment of industrial drainage. Recently, the governorate has begun a mechanical treatment operation on the El-Bats drain as the first step toward lake rehabilitation. This method includes the redirection of the El-Bats drain from immediately discharging wastewater into the lake to a new canal roughly 5 km long. Moving wastewater to a greater distance allows suspended particles and solid wastes to settle and increases oxygen dissolution, allowing more biodegradable organic matter to oxidize. After that, the water is subject to extra artificial aeration before passing through sand and gravel filters for further purification. This approach will require ongoing monitoring and laboratory analysis to determine its efficacy in decreasing pollution from the lake.

The mathematical techniques applied to compute the PLI, RI, and Dc in sediments with high efficiency and accuracy [66,78,80] are time-consuming. They require numerous mathematical steps to convert enormous metals data into a single value to describe the sediments' pollution levels. On the other hand, both PLSR and MLR procedures are simple to use and do not require multiple steps to compute the PLI, RI, and Dc of sediments with high performance. The multivariate regression models, such as PLSR and MLR, have recently been used as alternative methods to predict the environmental pollution indices based on data for concentrations of several metals [58,59,112,113]. To the best of our knowledge, the issue of predicting the PLI, RI, and Dc of sediments of Qaroun Lake using PLSR and MLR methods, based on several elements, has not yet been addressed.

The Cal. and Val. models presented the best performance to predict the PLI, RI, and Dc, based on selected 21 elements, with $R^2 = 0.948–0.991$ for PLSR, and with $R^2 = 0.760–0.998$ for MLR. In agreement with our results, recently, Abowaly et al. [114] found that the PLSR and MLR models performed best in predicting the PLI of the soil depending on data for the four investigated elements. With $R^2 = 0.89–0.93$ in the surface layer, 0.91–0.96 in the subsurface layer, 0.89–0.94 in the lowest layers, and 0.92–0.94 throughout the three layers, the validation (Val.) models fared the best in predicting the PLI. Finally, the findings of this study demonstrate that two approaches, PLSR and MLR, can assess PLI, RI, and Dc in sediments of Qaroun Lake.

## 5. Conclusions

The current study might be regarded as one of the Qaroun Lake geo-environmental monitoring activities. The concentrations for all examined elements reflect the highest values in the organic-rich and clayey sediments of the eastern and middle parts of the lake. Meanwhile, the lowest concentration values generally were observed in the western and northwest sediment samples, characterized by light colours reflecting less organic matter and clay mineral percentages. The distribution of metals in lake bottom sediment may be controlled by: (1) percentages of organic matter; (2) clay minerals; (3) coagulations result from the mix of fresh water with saline water; (4) salinity; (5) iron and magnesium oxides; (6) oxidation-reduction processes; (7) metal sources. The high levels of Al and Fe in the sediments studied suggested spontaneous contamination by terrigenous material. The tested lake sediments demonstrated very high contamination levels of As, Cd, Sb, Se, Ga, and Zr. Excluding the Al normalized element, all the investigated metals have anthropogenic sources, and Mg was the only investigated element that has a natural origin. Qaroun Lake sediments represent a very high level of contamination (Dc > 32). The eastern and southern sides of the middle part of the lake were polluted (PLI > 1), and the western and west northern parts were still unpolluted (PLI < 1). The bottom sediment of Qaroun Lake was shown to be at very high ecological risk. The tested metals positively correlated with Al, Fe, and Mn, and negatively linked with As, Se, and P, indicating that these metals may have different sources and distributions. The metals in Qaroun Lake sediment may originate from a mixed source (natural and anthropogenic). The potential environmental toxic metals As, Se, Cd, Zr, Hf, Ga, Sb, Cr, Ni, and P in the bottom sediment samples

of Qaroun Lake mainly originated from anthropogenic activity. Based on the preceding, it is evident that elements contaminated the Qaroun Lake sediments, and the lake will become unsuitable for recreational activities or fishing due to the harm present to human health and the environment. As a result, substantial and necessary steps must be taken to regulate sewage entrance, treat it before it enters the Lake, and manage the lake water quality and sediments.

In calibration and validation data sets, the PLSR and MLR models performed well in estimating the Dc, PLI, and RI of sediments, with the highest $R^2$ values, lowest RMSE and MAD values, and highest slope values. Future research should be concerned with studying the stability of PLSR and MLR models to assess Dc, PLI, and RI or other pollution indices of sediments under various environmental conditions.

**Supplementary Materials:** The following are available online at https://www.mdpi.com/article/10.3390/jmse9121443/s1, Table S1. The average value of metal concentrations in Qaroun Lake sediments in the year 2018; Table S2. The average value of metal concentrations in Qaroun Lake sediments in the year 2019.

**Author Contributions:** Conceptualization, A.H.S. and M.M.A.E.-S.; methodology, A.H.S., M.M.A.E.-S., F.S.M., M.G., M.M.K. and S.E.; software, A.H.S., M.G., S.E., A.M.G. and M.M.A.E.-S.; validation, A.H.S., S.E., S.D. and M.M.A.E.-S.; formal analysis, A.H.S., M.G., S.E., R.D. and M.M.A.E.-S.; investigation A.H.S., M.G., S.E., M.E.M. and M.M.A.E.-S.; resources, F.S.M. and M.M.A.E.-S.; data duration, A.H.S., M.G., M.M.K., S.E., F.S.M. and M.M.A.E.-S.; writing—original draft preparation, A.H.S., M.G., S.E. and M.M.A.E.-S.; writing—review and editing, A.H.S. and S.E.; supervision, A.M.G., S.D. and R.D. project administration, A.M.G., S.D. and R.D. funding acquisition, A.M.G., S.D. and R.D. All authors have read and agreed to the published version of the manuscript.

**Funding:** This research received no external funding.

**Data Availability Statement:** Data are contained within the article.

**Acknowledgments:** The authors would like to thank Anas A. Abouelkhair at the University Central Laboratory, Prince Sattam bin Abdulaziz University and Theodore Danso Marfo Department of Environmental Management Technology Cape Coast Technical University, Cape Coast—Ghana for their kind assistance, help during this work.

**Conflicts of Interest:** The authors declare no conflict of interest.

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
