# Peer review of "Environmental Pollution Indices and Multivariate Modeling Approaches for Assessing the Potentially Harmful Elements in Bottom Sediments of Qaroun Lake, Egypt"

_jmse, doi:10.3390/jmse9121443_

Round 1

Reviewer 1 Report

The work seems to be interesting by goal and design. The obtained results were well interpreted and discussed. The validity of obtained data was also proved through the statistical methodology. Some improvements are required before publication.

- Conclusion:  I propose to shorten the conclusions a bit. The leading general information lines 702-706 can be deleted. The conclusion should be concise within 1 paragraph.

Author Response

  • The work seems to be interesting by goal and design. The obtained results were well interpreted and discussed. The validity of obtained data was also proved through the statistical methodology.

Response: We greatly appreciate your critical observations as well as your constructive and helpful comments. We hope that we could address your questions/comments by the explanations and revisions made in the manuscript. We believe that the manuscript is substantially improved after making the suggested revisions.

  • Conclusion:  I propose to shorten the conclusions a bit. The leading general information lines 702-706 can be deleted. The conclusion should be concise within 1 paragraph.

Response: Thank you for your suggestion. We deleted the lines 702-706 and the conclusion was shorted as possible.

Reviewer 2 Report

This a well structured case study indicating comprehensively the estimation of the Dc, PLI and RI of sediments using the well known PLSR and MLR models. However, major changes in the paper are recommended before its acceptance for publication.

1. The bathymetry of the lake should be included to identify those areas that major accumulations due to bottom variations could be observed.
2. Part of the lake is characterised as shallow with depths <3m.  In contrast to deep water bodies, shallow systems are often characterized by a greater littoral zone and a closer contact with the surrounding lands, entailing a stronger aquatic-terrestrial exchange of matter and organisms. The water volume is relatively low. Together with polymixis, this results in enhanced benthic-pelagic coupling and greater impact of sediment processes in the water column. That means that the effect of sediments resuspension in water column should be assessed, while in the areas of more than 5m a possible long last thermal stratification in summer should be also discussed.

Author Response

  • This well-structured cases study indicating comprehensively the estimation of the Dc, PLI and RI of sediments using the well-known PLSR and MLR models.

Response: We greatly appreciate your critical observations as well as your constructive and helpful comments. We hope that we could address your questions/comments by the explanations and revisions made in the manuscript. We believe that the manuscript is substantially improved after making the suggested revisions.

  • The bathymetry of the lake should be included to identify those areas that major accumulations due to bottom variations could be observed.

Response: Thank you for your comment. The bathymetry of lake ranged from 2 to 5 m, and samples collected in this rang (Material and method section, from line 165 to 166). 

  • Part of the lake is characterized as shallow with depths <3m. In contrast to deep water bodies, shallow systems are often characterized by a greater littoral zone and a closer contact with the surrounding lands, entailing a stronger aquatic-terrestrial exchange of matter and organisms. The water volume is relatively low. Together with polymixis, this results in enhanced benthic-pelagic coupling and greater impact of sediment processes in the water column. That means that the effect of sediments resuspension in water column should be assessed, while in the areas of more than 5m a possible long last thermal stratification in summer should be also discussed.

Response: Thank you for your suggestion. The studied sediments are bottom sediment and it is not suspended sediment as illustrated in material and methods section. As well as the collected samples was obtained from the range of 2 to 5 m depth.

Reviewer 3 Report

Overall comments

The manuscript requires extensive language editing. The abstract for example needs a complete rewrite.

Throughout the work, the authors talk of “sediments” but do not indicate if these are suspended sediment particles or bottom sediments. This needs to be clear. It is even gets confusing with a phrase like “surface bottom sediment” (Line 97; 185).

For sediment collection, the authors should mention whether they collected core samples or grab samples. Also, how deep below the surface were the sediments collected?

In the materials and methods and discussion, it would be good to mention something on the geology of the area. Metals in the environment could also arise from natural dissolution. The authors have mention sand dunes as one of the natural sources, but a brief description of the geology would indicate the extent of the anthropogenic activities.

The authors also mentioned that particle size mattered in the accumulation of heavy metal pollutants in sediments (Line 562). Also, the authors used a 2.00 mm sieve for processing. It would therefore be good to briefly describe the percentage particle size distribution of the sediments to understand the rate of accumulation.

Specific comments

Line 32-41: “Through…..principal component analysis (PCA)..” Hanging sentences. Please rephrase. The sentences structures do not bring out the message. E.g. …. “affecting the lake environmental system..” should just be “the lake system”

Line 43: Delete “by”

Line 45: Change “have” to “had”

Line 45-46: “The results of Dc revealed to all samples are a very high degree of contamination”. Not clear what the authors are saying here.

Line 62: The lakes' aquatic environment? I don’t think the lake has a terrestrial environment. The lake is already an aquatic environment, so please rephrase. The whole sentence needs to be rephrased.

Figure 1: Define 1, 2, 3,…in the map legend

Line 206: Delete “value”

Table 2: Please provide the unit of measurement

Figure 4: Include description of the figure in the legend. What do the different colours and circle sizes mean?

Author Response

Reviewer #3:

  • We greatly appreciate your critical observations as well as your constructive and helpful comments. We hope that we could address your questions/comments by the explanations and revisions made in the manuscript. We believe that the manuscript is substantially improved after making the suggested revisions.
  • The manuscript requires extensive language editing. The abstract for example needs a complete rewrite

  • Response: Thank you for this comment. English editing was improved across the manuscript as you suggested. As well as, it is possible for sending it to the language office in MDPI, if it is necessary after this modification.  The abstract also was improved .
  • Throughout the work, the authors talk of “sediments” but do not indicate if these are suspended sediment particles or bottom sediments. This needs to be clear. It is even gets confusing with a phrase like “surface bottom sediment” (Line 97; 185).

Response: Thank you for your critical observations. The collected sediments were bottom sediments, and the word “surface” was deleted from the  all manuscript.   

  • For sediment collection, the authors should mention whether they collected core samples or grab samples. Also, how deep below the surface were the sediments collected?

Response: Thank you for your comment. The samples were collected by grab, and collected samples were obtained from depths ranges between 2 to 5 meters. (Material and method section, line 197 to line 200).    

  • In the materials and methods and discussion, it would be good to mention something on the geology of the area. Metals in the environment could also arise from natural dissolution. The authors have mention sand dunes as one of the natural sources, but a brief description of the geology would indicate the extent of the anthropogenic activities.

Response: Thank you for your suggestion. The geology of the study area was added as you suggested. (Material and method section, from lines 181 to 187).  

  • The authors also mentioned that particle size mattered in the accumulation of heavy metal pollutants in sediments (Line 562). Also, the authors used a 2.00 mm sieve for processing. It would therefore be good to briefly describe the percentage particle size distribution of the sediments to understand the rate of accumulation.
  •  

Response: Thank you for your suggestion. We have the unpublished data which were referred to the sediments of the eastern and southern sides of the Lake with fine fractions of about > 70 % less than 63 microns (Line 207 to 208).

  • Line 32-41: “Through…..principal component analysis (PCA)..” Hanging sentences. Please rephrase. The sentences structures do not bring out the message. E.g. …. “affecting the lake environmental system..” should just be “the lake system”.

Response: Thank you for your suggestion. We rephrase the sentences as you suggested. (Abstract section, from line 32 to 38).

  • Line 43: Delete “by”

Response: Thank you for your suggestion. The word “by” was deleted (Line 42). 

  • Line 45: Change “have” to “had”

Response: Thank you for your suggestion. The word “have” was changed to “had” (Line 44). 

  • Line 45-46: “The results of Dc revealed to all samples are a very high degree of contamination”. Not clear what the authors are saying here.

Response: Thank you for your comment. According to the results of Dc, all collected samples, were categorized under a very high degree of contamination. (Abstract section, lines 45 to 46).

  • Line 62: The lakes' aquatic environment? I don’t think the lake has a terrestrial environment. The lake is already an aquatic environment, so please rephrase. The whole sentence needs to be rephrased.

Response: Thank you for your critical observations. We rephrase the sentence according to your suggestion. (Introduction section, lines 62 to 64).

  • Figure 1: Define 1, 2, 3,…in the map legend

Response: Thank you for your critical observations. The number of the collected samples was defined in the legend of the map as you suggested.

  • Line 206: Delete “value”

Response: Thank you for your critical observations. The word “value” was deleted. (Sampling and analyses section, line 219).

  • Table 2: Please provide the unit of measurement

Response: Thank you for your critical observations. The metal concentrations in ppm, except for aluminum, iron, and zirconium are in %. ( in the title of Tale).

  • Figure 4: Include description of the figure in the legend. What do the different colours and circle sizes mean?

  • Response: Thank you for your critical observations. For color, blue color means positive correlation and red color means negative correlation. For size of circle refers to the degree of correlation. It was written under the table 4.

Round 2

Reviewer 2 Report

The paper is significantly improved and can be accepted for publication.

Author Response

  • The paper is significantly improved and can be accepted for publication.

Response: We greatly appreciate your critical observations as well as your constructive and helpful comments.

Reviewer 3 Report

The scientific comments have been addressed but the language has not improved. I suggest the authors us the service of a professional language editor. For example, Line 37-40 has two hanging sentences. The same applies for Line 60-61, 65-66, and many other instances.

Author Response

Reviewer #3:

  • The scientific comments have been addressed but the language has not improved. I suggest the authors us the service of a professional language editor. For example, Line 37-40 has two hanging sentences. The same applies for Line 60-61, 65-66, and many other instances.

Response: We greatly appreciate your comments as well as your constructive. The English language of our manuscript was edited by Native English speaker. Please find attached letter from Native English speaker.
